# Skyformer: Remodel Self-Attention with Gaussian Kernel and Nyström Method

**Yifan Chen**,* **Qi Zeng**,* **Heng Ji,  Yun Yang**
University of Illinois Urbana-Champaign
`{yifanc10, qizeng2, hengji, yy84}@illinois.edu`

## Abstract

Transformers are expensive to train due to the quadratic time and space complexity in the self-attention mechanism. On the other hand, although kernel machines suffer from the same computation bottleneck in pairwise dot products, several approximation schemes have been successfully incorporated to considerably reduce their computational cost without sacrificing too much accuracy. In this work, we leverage the computation methods for kernel machines to alleviate the high computational cost and introduce Skyformer, which replaces the softmax structure with a Gaussian kernel to stabilize the model training and adapts the Nyström method to a non-positive semidefinite matrix to accelerate the computation. We further conduct theoretical analysis by showing that the matrix approximation error of our proposed method is small in the spectral norm. Experiments on Long Range Arena benchmark show that the proposed method is sufficient in getting comparable or even better performance than the full self-attention while requiring fewer computation resources.

## 1   Introduction

The cost of language model training increases exponentially. Among different models, Transformer-based language models [Vaswani et al., 2017, Devlin et al., 2019, Liu et al., 2019, Lewis et al., 2020] are shown to enjoy state-of-the-art (SOTA) performances on many Natural Language Processing (NLP) tasks despite their enormous training cost. One of the computation bottlenecks lies in the self-attention mechanism, which is known to be resource-intensive with quadratic time and space complexity ($O(n)$ where $n$ is the input sequence length). Consequently, Transformers cannot support long sequence processing and large batch size with limited resources.

The challenge of improving computational efficiency of Transformers has motivated several recent studies on attention acceleration, using either sparse attention pattern [Qiu et al., 2020, Child et al., 2019, Zaheer et al., 2020, Beltagy et al., 2020, Kitaev et al., 2020] or low-rank approximation [Choromanski et al., 2020, Wang et al., 2020]. However, there is usually a lack of theoretical analysis on the approximation error of these methods due to the complex softmax structure, which makes the theoretical comparison between the efficiency of each method infeasible. It is also unclear in theory how to set the hyper-parameters of those methods to attain a desired level of approximation accuracy.

Another issue of Transformers is the training instability that small perturbations in parameter updates tend to be amplified, resulting in significant disturbances in the model output [Liu et al., 2020a]. Transformers on some NLP tasks have shown to be sensitive to hyper-parameters, learning schedulers, or even random seeds, which usually demands a time-costly grid search for the best configuration in real-world applications. It has also been observed in our experiments that a slight change in the learning rate may cause the failure of convergence for some models. We conjecture that the instability

---

*   * Equal contribution.

35th Conference on Neural Information Processing Systems (NeurIPS 2021).

in Transformer training comes from the softmax structure, as the un-normalized attention score matrices before softmax tend to have extremely large condition numbers due to its fast singular value decay.

To alleviate the instability issue, an extra factor of $1/\sqrt{p}$ in the softmax kernel SM is suggested by Vaswani et al. [2017] to restrain the scale variation; Liu et al. [2020a] proposes a new scheme to control the magnitude of output change and stabilize the training in early stages. In practice, we also need to consider the lower numerical precision of GPU implementation in model training, which further deteriorates the stability.

Kernel methods may be the answer to both challenges. As pointed out by Choromanski et al. [2020], the softmax structure is closely related to Gaussian kernels up to diagonal matrix multiplications, as the pairwise dot products naturally appear when expanding the squared $\ell_2$ distance. We further notice some important connections between self-attention and Gaussian kernels. First, the un-normalized attention score matrix can be formed via basic matrix operations on an empirical Gaussian kernel matrix. Moreover, the form of Gaussian kernels has the natural interpretation of assigning "attention" to different tokens. Compared to the softmax function, Gaussian kernels automatically perform the normalization as softmax does (c.f. Section 4.1). These observations motivate us to replace the softmax structure with Gaussian kernels. As we demonstrated in this paper, the new attention model, **Kernelized Attention**, empirically stabilizes the model training while being comparable to self-attention in model accuracy.

To further improve the efficiency, we propose **Skyformer** (**S**ymmetrization of **K**ernelized attention for N**Y**ström method) to accelerate kernelized attention. Skyformer adapts the Nyström method [Williams and Seeger, 2001, Drineas et al., 2005] to the non-PSD empirical Gaussian kernel matrix (as query matrices in general do not equal to key matrices), by instead lifting the kernelized attention score matrix into a large PSD matrix that contains the un-normalized attention score matrix as the off-diagonal block. We further conduct theoretical analysis by showing that Skyformer has a small matrix approximation error on kernelized attention in the spectral norm. Our experiments on the LRA benchmark show that Skyformer consistently uses less space and time while achieving better accuracy than other baseline methods.

In summary, our main contributions are:

(1) We revisit the intrinsic connection between self-attention and kernel methods, and explore a new kernel-based structure, kernelized attention, to stabilize the training of Transformers.

(2) We propose Skyformer, which approximates the kernelized attention via low dimensional randomized sketches by adapting the Nyström method to a non-PSD matrix. We provide the theoretical guarantee that the matrix multiplication error is small in term of spectral norm.

(3) Extensive experiments show that Skyformer achieves comparable performance to the original self-attention with fewer computational costs.[2]

## 2   Related Work

Among all the transformer acceleration methods, including attention layer simplification by pruning redundant attention heads [Voita et al., 2019, Michel et al., 2019] and model size reduction with knowledge distillation [Jiao et al., 2020, Tang et al., 2019, Liu et al., 2020b], we focus on attention approximation models, which are closely related to kernel methods.

To reduce the time and space complexity by avoiding exhaustive computation over the attention metric, recent studies propose to apply sparse attention patterns to limit the numbers of elements participating in matrix multiplications [Qiu et al., 2020, Child et al., 2019, Zaheer et al., 2020, Beltagy et al., 2020]. Beyond limiting the attention to fixed patterns, some approaches learn the patterns by determining token assignments to relevant groups [Kitaev et al., 2020, Roy et al., 2021]. Those models utilize local and global information in the attention score matrix to perform approximation, which coincides with the attempt to accelerate the computation in Gaussian processes [Snelson and Ghahramani, 2007].

---

[2]Our code is released at `https://github.com/pkuzengqi/Skyformer`

The attention score matrix is known to exhibit a very fast rate of singular value decay [Bhojanapalli et al., 2020, Dong et al., 2021], similar to that of an empirical kernel matrix [Yang et al., 2017]. This near singular property motivates many low-rank attention approximation methods to skillfully leverage the computation techniques in kernel methods. Among them, Linformer [Wang et al., 2020] compresses the size of the key and value matrix with random projections based on the Johnson–Lindenstrauss transform, a common randomized sketching method in Gaussian processes [Yang et al., 2017]; Reformer [Kitaev et al., 2020] applies locality-sensitive hashing (LSH) [Har-Peled et al., 2012] to simplify the computation of the attention score matrix, which is widely used in kernel density estimation [Charikar and Siminelakis, 2017, Backurs et al., 2019]; Performer [Choromanski et al., 2020] projects both query and key matrix through random Fourier features [Rahimi et al., 2007], heavily exploiting Bochner Theorem for stationary kernels.

The most related papers to ours are linear attention [Katharopoulos et al., 2020], Synthesizer [Tay et al., 2020a], and Nyströmformer [Xiong et al., 2021]. Linear attention takes the softmax structure in self-attention as a measure of similarity and replaces it with the dot product of separately activated query and key matrices; Synthesizer aims to modify the original self-attention by replacing the dot product before softmax with Synthetic Attention, which generates the alignment matrix independent of token-token dependencies. Their attempts indicate that the softmax structure in self-attention is not the only feasible choice, and justify our usage of kernelized attention. Rather than remodeling self-attention, Nyströmformer applies the Nyström method [Williams and Seeger, 2001, Drineas et al., 2005], a powerful and effective method for large-scale kernel machines acceleration, to approximate the attention score matrix. However, Nyströmformer applies the Nyström method to a non-PSD matrix, and thus fails to utilize the full potential of the Nyström method. This issue is resolved in our proposed Skyformer by instead lifting the kernelized attention score matrix into a large PSD matrix which contains the target non-PSD matrix as its off-diagonal block. For more details on attention approximation methods, we refer readers to a survey paper on efficient transformers [Tay et al., 2020c].

## 3 Preliminaries and notations

### 3.1 Revisiting self-attention

For a given input sequence $\boldsymbol{X} \in \mathbb{R}^{n \times d_0}$ of length $n$ and embedding dimension $d_0$, The dot-product attention for a single head in Transformer Vaswani et al. [2017] is defined as

$$\text{Attention}(\boldsymbol{Q}, \boldsymbol{K}, \boldsymbol{V}) = \text{softmax}\left(\frac{\boldsymbol{Q}\boldsymbol{K}^T}{\sqrt{p}}\right)\boldsymbol{V}$$

where $\boldsymbol{Q} = \boldsymbol{X}\boldsymbol{W}_Q$, $\boldsymbol{K} = \boldsymbol{X}\boldsymbol{W}_K$, and $\boldsymbol{V} = \boldsymbol{X}\boldsymbol{W}_V$, and $\boldsymbol{W}_Q, \boldsymbol{W}_K$ and $\boldsymbol{W}_V$ are the query, key, and value weight metrics that linearly project the input $\boldsymbol{X}$ of $d_0$ dimension to an output tensor of $p$ dimensions.

To simplify the future analysis, the left softmax term can be rewritten into $\boldsymbol{D}^{-1}\boldsymbol{A}$, where $\boldsymbol{A} := \exp(\boldsymbol{Q}\boldsymbol{K}^T/\sqrt{p})$ is the un-normalized attention score matrix; $\boldsymbol{D}$ is a diagonal matrix whose diagonal is $\exp(\boldsymbol{Q}\boldsymbol{K}^T/\sqrt{p}) \cdot \mathbf{1}$ (by convention $\mathbf{1}$ is a size-$n$ vector with all elements being 1). Following the notation in Performer [Choromanski et al., 2020], we define $\text{SM}(\boldsymbol{q}, \boldsymbol{k}) := \exp(\boldsymbol{q}^T\boldsymbol{k}/\sqrt{p})$ as the softmax kernel function, and represent $\boldsymbol{A}$ by the notation $\text{SM}(\boldsymbol{Q}, \boldsymbol{K})$, which means the element $a_{ij}$ from the $i$-th row and $j$-th column in $\boldsymbol{A}$ is equal to $\text{SM}(\boldsymbol{q}_i, \boldsymbol{k}_j)$. Throughout this paper $\boldsymbol{q}_i$ (resp. $\boldsymbol{k}_j$) means the $i$-th (resp. $j$-th) row in $\boldsymbol{Q}$ (resp. $\boldsymbol{K}$).

We close this subsection with a short lemma to show $\text{SM}(\cdot, \cdot)$ is a positive semidefinite (PSD) kernel function [Wainwright, 2019, Definition 12.6] by relating it to Gaussian kernels.

**Lemma 1.** *$SM(\cdot, \cdot)$ is a PSD kernel function. Equivalently, for all integers $n \geq 1$ and elements $\{\boldsymbol{q}_i\}_{i=1}^n \subseteq \mathbb{R}^p$, the n-by-n matrix $\boldsymbol{C} = SM(\boldsymbol{Q}, \boldsymbol{Q})$ is PSD.*

*Proof.* We first state an important equation to connect the softmax kernel and Gaussian kernels as follows:

$$\text{SM}(\boldsymbol{q}_i, \boldsymbol{q}_j) = \exp\left(\frac{\boldsymbol{q}_i^T\boldsymbol{q}_j}{\sqrt{p}}\right) = \exp\left(\frac{\|\boldsymbol{q}_i\|^2}{2\sqrt{p}}\right)\exp\left(-\frac{\|\boldsymbol{q}_i - \boldsymbol{q}_j\|^2}{2\sqrt{p}}\right)\exp\left(\frac{\|\boldsymbol{q}_j\|^2}{2\sqrt{p}}\right).$$

The middle part $\exp\left(\frac{\|\boldsymbol{q}_i - \boldsymbol{q}_j\|^2}{2\sqrt{p}}\right)$ is exactly a Gaussian kernel with bandwidth $p^{\frac{1}{4}}$. (Choromanski et al. [2020] have more discussion on the findings.)

Through this equation, we can rewrite $\boldsymbol{C}$ as

$$\boldsymbol{C} = \boldsymbol{D}_Q^{1/2} \cdot \kappa\left(\frac{\boldsymbol{Q}}{p^{1/4}}, \frac{\boldsymbol{Q}}{p^{1/4}}\right) \cdot \boldsymbol{D}_Q^{1/2}, \tag{1}$$

where $\boldsymbol{D}_Q$ is a diagonal matrix with elements $(\boldsymbol{D}_Q)_{ii} = \exp\left(\frac{\|\boldsymbol{q}_i\|^2}{\sqrt{p}}\right), \forall i \in [n]$, and $\kappa(\boldsymbol{q}_i, \boldsymbol{q}_j) := \exp\left(-\|\boldsymbol{q}_i - \boldsymbol{q}_j\|^2/2\right)$ is the standard Gaussian kernel function.

We prove the lemma by using the fact that $\kappa$ is a PSD kernel and $\kappa\left(\frac{\boldsymbol{Q}}{p^{1/4}}, \frac{\boldsymbol{Q}}{p^{1/4}}\right)$ is a PSD matrix. $\diamondsuit$

## 3.2 Nyström method

Due to the intrinsic low-rankness of an empirical kernel matrix $\boldsymbol{B}$, the so-called Nyström method that replaces $\boldsymbol{B}$ with its low-rank approximation $\tilde{\boldsymbol{B}}$, has been applied to accelerate kernel methods [Gittens and Mahoney, 2016, Kumar et al., 2009, Williams and Seeger, 2001]. Specifically, the Nyström approximation of $\boldsymbol{B}$ is the matrix $\tilde{\boldsymbol{B}} = \boldsymbol{B}\boldsymbol{S}(\boldsymbol{S}^T\boldsymbol{B}\boldsymbol{S})^\dagger\boldsymbol{S}^T\boldsymbol{B}$, where $(\cdot)^\dagger$ denotes the Moore-Penrose pseudoinverse of a matrix, and $\boldsymbol{S} \in \mathbb{R}^{n \times d}$ is a zero-one sub-sampling matrix whose columns are a subset of the columns in $\boldsymbol{I}$, indicating which $d$ observations have been selected. The formal definition of the uniform sub-sampling matrix is given as follows:

**Definition 1** (Uniform sub-sampling matrix). *For a random matrix $\boldsymbol{S} \in \mathbb{R}^{n \times d}$, if $\boldsymbol{S}$ has i.i.d. columns and the $j$-th column $\boldsymbol{S}^{(j)}$ can randomly be $\sqrt{\frac{1}{d}}\boldsymbol{e}_i$ with probability $\frac{1}{n}$, where $\boldsymbol{e}_i$ is the $i$-th column of the $n$-by-$n$ identity matrix $\boldsymbol{I}_n$, then $\boldsymbol{S}$ is called a uniform sub-sampling.*

We close this subsection with a remark that it is not appropriate to directly extend the Nyström method from kernel method to self-attention due to a core requirement that $\boldsymbol{B}$ should be PSD with consideration of approximation performance improvement. We will show in the next section how to address this challenge and properly adapt Nyström method to non-PSD matrices.

## 3.3 Approximation evaluation

Beyond the time and space complexity, attention acceleration methods have been mostly evaluated with empirical experiment results, such as the perplexity of pretrained language models and the fine-tuned performance on downstream natural language understanding tasks. Specifically, Long Range Arena benchmark [Tay et al., 2020b] has been proposed to systematically evaluate the performance of efficient transformers with ten NLP tasks in long-context scenarios. However, such empirical results are indirect for theoretical analysis. Therefore, we introduce a common criterion used in matrix approximation, spectral norm, to ease the future discussion on performance.

**Definition 2** (Spectral norm guarantee for matrix approximation (MA)). *Given a matrix $\boldsymbol{M} \in \mathbb{R}^{n_1 \times n_2}$, two constants $\varepsilon > 0, \delta < \frac{1}{2}$, we say that its approximation matrix $\widetilde{\boldsymbol{M}} \in \mathbb{R}^{n_1 \times n_2}$ satisfies $(\varepsilon, \delta)$-MA property for $\boldsymbol{M}$, if*

$$\mathbb{P}\left\{\|\boldsymbol{M} - \widetilde{\boldsymbol{M}}\| > \varepsilon\|\boldsymbol{M}\|\right\} < \delta. \tag{2}$$

In previous works, the direct analysis of the approximation error to the entire output $\boldsymbol{D}^{-1}\boldsymbol{A}\boldsymbol{V}$ in the $(\varepsilon, \delta)$-MA manner is usually spared due to the difficulty caused by the complex softmax structure. In this paper, with the new kernelized attention, we are allowed to perform the analysis through the existing theoretical results in kernel methods. Consequently, in Section 4.5 we are able to give a relatively precise error analysis on the approximation of Skyformer to the entire kernelized attention, which eases the future comparison with other methods approximating kernelized attention.

# 4 Method

## 4.1 Kernelized Attention

Kernelized Attention replaces the softmax structure in vanilla self-attention with a Gaussian kernel, and the new attention model is stated as:

$$\text{Kernelized-Attention}(\boldsymbol{Q}, \boldsymbol{K}, \boldsymbol{V}) = \boldsymbol{CV} := \kappa \left( \frac{\boldsymbol{Q}}{p^{1/4}}, \frac{\boldsymbol{K}}{p^{1/4}} \right) \boldsymbol{V}, \tag{3}$$

where we define the $n$-by-$n$ matrix $\boldsymbol{C}$ as the kernelized attention score matrix $\kappa(\boldsymbol{Q}/p^{1/4}, \boldsymbol{K}/p^{1/4})$.

The justification for using the kernelized attention model is as follows. A significant advantage of softmax attention is that tokens are allowed to attend to a limited number of other important tokens in the sequence. We observe that Gaussian kernel function can play a similar role. The expression of a Gaussian kernel is $\kappa(\boldsymbol{q}_i, \boldsymbol{k}_j) := \exp\left(-\|\boldsymbol{q}_i - \boldsymbol{k}_j\|^2/2\right)$. Via this expression, for token $i$ in the query, Gaussian kernel assigns a large attention score to the token $j$ when $\boldsymbol{k}_j$ is close to $\boldsymbol{q}_i$. The distance-based weight assignment is indeed considered as a major reason why kernel methods are powerful. The form of kernelized attention also leads to an automatic normalization. Based on Equation (1), the new attention model can be rewritten in terms of the un-normalized attention score matrix $\boldsymbol{A}$ as

$$\text{Kernelized-Attention}(\boldsymbol{Q}, \boldsymbol{K}, \boldsymbol{V}) = \left( \boldsymbol{D}_Q^{-1/2} \cdot \boldsymbol{A} \cdot \boldsymbol{D}_K^{-1/2} \right) \boldsymbol{V},$$

where $\boldsymbol{D}_Q$ (resp. $\boldsymbol{D}_K$) is a diagonal matrix with elements $(\boldsymbol{D}_Q)_{ii} = \exp\left(\frac{\|\boldsymbol{q}_i\|^2}{\sqrt{p}}\right)$ (resp. $(\boldsymbol{D}_K)_{ii} = \exp\left(\frac{\|\boldsymbol{k}_i\|^2}{\sqrt{p}}\right)$), $\forall i \in [n]$. We remark the kernelized attention model can thus be formally taken as a variant of the original self-attention, which instead normalizes the matrix $\boldsymbol{A}$ in a form of $\boldsymbol{D}^{-1}\boldsymbol{A}$. The intrinsic normalization allows kernelized attention to have a more reasonable condition number than self-attention, which benefits the stability of model training. To demonstrate the improvement in stability, we additionally provide a toy experiment in Appendix F, which shows the "condition number" of kernelized attention is smaller than self-attention. Moreover, empirical evaluation in Section 5 supports our claim that the new attention model can attain a comparable performance to the original attention model.

## 4.2 Skyformer: a modified Nyström method

Before jumping into details of Skyformer, we first propose a method to apply Nyström method to approximate an asymmetric (and thus non-PSD) empirical kernel matrix $\boldsymbol{B}$ constructed with any PSD kernel $\phi(\cdot, \cdot)$. Specifically, with two different $n$-by-$p$ design matrices $\boldsymbol{Q}$ and $\boldsymbol{K}$, its element $b_{ij}$ from the $i$-th row and $j$-th column in $\boldsymbol{B}$ is equal to $\phi(\boldsymbol{q}_i, \boldsymbol{k}_j)$, where $\boldsymbol{q}_i$ (resp. $\boldsymbol{k}_j$) is the $i$-th (resp. $j$-th) row in $\boldsymbol{Q}$ (resp. $\boldsymbol{K}$). We remark this type of empirical kernel matrices involves the un-normalized attention score matrix $\boldsymbol{A} := \text{SM}(\boldsymbol{Q}, \boldsymbol{K})$, and the empirical Gaussian kernel matrix $\boldsymbol{C} := \kappa(\boldsymbol{Q}/p^{1/4}, \boldsymbol{K}/p^{1/4})$. Therefore this method leads to a low-rank approximation to the output of either self-attention $\boldsymbol{D}^{-1}\boldsymbol{AV}$ or Kernelized Attention $\boldsymbol{CV}$. ($\boldsymbol{D}$ in self-attention can be obtained by computing $\boldsymbol{A} \cdot \mathbf{1}$, and thus a low-rank approximation to $\boldsymbol{A}$ also implies an approximation to $\boldsymbol{D}$.)

Computational details are stated as follows. To tackle the challenge of approximating a non-PSD matrix $\boldsymbol{B}$, our first step is to complete the matrix into a PSD matrix $\bar{\boldsymbol{B}}$:

$$\bar{\boldsymbol{B}} := \phi\left( \begin{pmatrix} \boldsymbol{Q} \\ \boldsymbol{K} \end{pmatrix}, \begin{pmatrix} \boldsymbol{Q} \\ \boldsymbol{K} \end{pmatrix} \right). \tag{4}$$

Then we approximate $\bar{\boldsymbol{B}}$ with $\tilde{\bar{\boldsymbol{B}}}$ through

$$\tilde{\bar{\boldsymbol{B}}} = \bar{\boldsymbol{B}}\mathbf{S}(\mathbf{S}^T\bar{\boldsymbol{B}}\mathbf{S})^\dagger\mathbf{S}^T\bar{\boldsymbol{B}}, \tag{5}$$

where $\boldsymbol{S}$ is a $2n$-by-$d$ uniform sub-sampling matrix as defined in Definition 1. The final approximation will be given as

$$\tilde{\boldsymbol{B}} := (\boldsymbol{I}, \mathbf{0})\tilde{\bar{\boldsymbol{B}}}(\mathbf{0}, \boldsymbol{I})^T. \tag{6}$$

The original matrix $\boldsymbol{B}$ can be well-approximated by $\tilde{\boldsymbol{B}}$ due to the following inequality

$$\|\boldsymbol{B} - \tilde{\boldsymbol{B}}\| = \|(\boldsymbol{I}, \boldsymbol{0})(\bar{\boldsymbol{B}} - \tilde{\bar{\boldsymbol{B}}})(\boldsymbol{0}, \boldsymbol{I})^T\| \leq \|\bar{\boldsymbol{B}} - \tilde{\bar{\boldsymbol{B}}}\|,$$

and thus we show our task of approximating the non-PSD matrix $\boldsymbol{B}$ boils down to well approximating the PSD matrix $\bar{\boldsymbol{B}}$.

**Remark.** The reason why $\boldsymbol{B}$ can be well approximated by a low-rank $\tilde{\boldsymbol{B}}$ is that as an empirical kernel matrix the eigenvalues in $\bar{\boldsymbol{B}}$ usually decay fast, and thus there are many small eigenvalues in the long tail. In this case, theoretically a low-rank matrix (e.g. truncated singular value decomposition (SVD) of $\bar{\boldsymbol{B}}$) has enough potential to well approximate the original matrix $\bar{\boldsymbol{B}}$ (and $\boldsymbol{B}$ accordingly) in terms of spectral norm.

With the derivation above, we officially introduce our proposed Skyformer as an approximation to Kernelized Attention, which applies the modified Nyström method to the kernelized attention score matrix $\boldsymbol{C}$. The next two subsections will continue our discussion on it, and respectively state the theoretical analysis of its approximation error and some details of its implementation in practice.

### 4.3 Error analysis of Skyformer

As mentioned, an implicit advantage of using Kernelized Attention is that we can leverage the existing conclusions for kernel methods to analyze the theoretical properties of the model. In this subsection, we aim to provide some theoretical analysis of its approximation error.

We state a high probability bound on the size $d$ of the sub-sampling matrix used in Skyformer to attain $(\varepsilon, \delta)$-MA property for the kernelized attention score matrix $\boldsymbol{C}$ by the following theorem. We refer the readers to the proof in Appendix D to take a closer look at our claim that the matrix to be approximated should be PSD is a key to the theoretical guarantee of Nyström method.

**Theorem 2** (Adapted from Lemma 9 and Theorem 3 [Musco and Musco, 2017]). *Consider the query, key, and value matrix $\boldsymbol{Q}, \boldsymbol{K}, \boldsymbol{V} \in \mathbb{R}^{n \times p}$ and two positive constants $\varepsilon < 1, \delta < \frac{1}{2}$. For the empirical Gaussian kernel matrix $\boldsymbol{C} := \kappa(\boldsymbol{Q}/p^{1/4}, \boldsymbol{K}/p^{1/4})$ defined above, we let $\lambda := \varepsilon\|\boldsymbol{C}\| < \|\bar{\boldsymbol{C}}\|$, where $\bar{\boldsymbol{C}}$ is the completion of $\boldsymbol{C}$ (similar to $\bar{\boldsymbol{B}}$, constructed as substituting the Gaussian kernel with bandwidth $p^{1/4}$ for the arbitrary kernel function $\phi$ in Equation (4)). We comment $\lambda$ serves as the regularization coefficient as well as the approximation error bound. To ease the analysis, we specifically define the $i$-th diagonal element of $\bar{\boldsymbol{C}}(\bar{\boldsymbol{C}} + \lambda\boldsymbol{I}_{2n})^{-1}$ as leverage score $\ell_i, \forall i = 1, \ldots, 2n$, and define their sum $\mathrm{Tr}\left(\bar{\boldsymbol{C}}(\bar{\boldsymbol{C}} + \lambda\boldsymbol{I})^{-1}\right)$ as the statistical dimension $d_{stat}$, which increases with $1/\varepsilon$ as $\lambda \propto \varepsilon$. Suppose $\boldsymbol{S}$ is a uniform sub-sampling matrix, and assume there exists a constant $\beta \in (0, 1]$ such that $\beta \leq \frac{d_{stat}}{2n\ell_i}, \forall i = 1, \ldots, 2n$. For the approximation matrix $\tilde{\bar{\boldsymbol{C}}}$ constructed with $\bar{\boldsymbol{C}}, \boldsymbol{S}$ as in Equation (5), there exists a constant $C$ such that if*

$$d \geq C\frac{d_{stat}}{\beta}\log\frac{n}{\delta}$$

*then $\tilde{\bar{\boldsymbol{C}}} \preccurlyeq \bar{\boldsymbol{C}} \preccurlyeq \tilde{\bar{\boldsymbol{C}}} + \lambda\boldsymbol{I}$ with probability $1 - \delta$. Here $\preccurlyeq$ denotes the Loewner ordering: $\boldsymbol{B} \preccurlyeq \boldsymbol{A}$ means $\boldsymbol{A} - \boldsymbol{B}$ is positive semidefinite. Furthermore, for our approximation $\tilde{\boldsymbol{C}}$ in Equation (6) to the kernelized attention score $\boldsymbol{C}$, we have*

$$\|\tilde{\boldsymbol{C}} - \boldsymbol{C}\| \leq \lambda = \varepsilon\|\boldsymbol{C}\|.$$

This theorem implies the time and space complexity of our proposed approximation depends on the statistical dimension $d_{stat}$. If we directly use the conclusion from Gaussian kernels, $d_{stat}$ should be $\widetilde{\mathcal{O}}(1)$ (complexity modulo poly-log term) [Yang et al., 2017] due to the exponential eigenvalue decay rate of Gaussian kernels, which is comparable to the complexity of most other efficient transformers. However, different than the case in the classical kernel methods, the distribution of the query and key matrix $\boldsymbol{Q}$ and $\boldsymbol{K}$ changes during the training procedure, which may invalidate the conclusion about $d_{stat}$. We leave the exact non-asymptotic analysis of the computational complexity for future work.

### 4.4 Workaround in implementation

A potential limitation with the implementation of the proposed method lies in the tricky fact that the matrix inversion on GPU is much slower and numerically less stable than the same operation on CPU

due to the different back-end libraries in the two platforms. We attempt to circumvent the problem by adapting the strategy in Nyströmformer [Xiong et al., 2021] to our setting. Specifically, we use the matrix-product-based iterative method [Razavi et al.] for finding approximate inverses, instead of some division-based methods (such as the conjugate gradient method) which induces some instability in model training.

To apply the iterative method and inverse matrix $\boldsymbol{M} = \boldsymbol{S}^T \bar{\boldsymbol{C}} \mathbf{S}$, we need to satisfy its assumption [Razavi et al., Theorem 2] that $\|\boldsymbol{I} - \boldsymbol{M}\| < 1$. In practice, we instead pass the matrix $\boldsymbol{D}_M^{-1/2}(\boldsymbol{M} + \gamma\boldsymbol{I})\boldsymbol{D}_M^{-1/2}$ as an input to the iterative method, where $\gamma > 0$ is a small constant and the diagonal matrix $\boldsymbol{D}_M$ is defined as $\mathrm{diag}\left((\boldsymbol{M} + \gamma\boldsymbol{I})\mathbf{1}\right)$. We give the following lemma to justify our practical usage of the method. The proof is deferred to Appendix E.

**Lemma 3.** *Given a constant $\gamma > 0$, if matrices $\boldsymbol{M}$ is constructed as $\boldsymbol{S}^T\bar{\boldsymbol{C}}\mathbf{S}$, and $\boldsymbol{D}_M$ are defined as above, then all the singular values of $\boldsymbol{D}_M^{-1/2}(\boldsymbol{M} + \gamma\boldsymbol{I})\boldsymbol{D}_M^{-1/2}$ are within $(0,1)$, which implies that $\|\boldsymbol{I} - \boldsymbol{D}_M^{-1/2}(\boldsymbol{M} + \gamma\boldsymbol{I})\boldsymbol{D}_M^{-1/2}\| < 1$.*

We further comment that numerically an implicit risk of the Schulz-type iterative method we use is the unintended consequence of "zero fill-in". If we use some sparse kernels (e.g. test functions with bounded support) other than Gaussian kernels, the empirical kernel matrices are sparse while the approximate inverse will converge to a dense matrix, which increases the computational cost.

### 4.5 Empirical approximation evaluation

Spectral norm, the maximum singular value of a matrix, is a computation-light indicator of matrix approximation performance. In this work, we compare the spectral norm of the difference between the outputs from attention functions and the output from vanilla self-attention with the same input.

We use the initialized and pretrained bert-base-cased models from Huggingface's implementation [Wolf et al., 2019] . The input vector $X$ is embedded from the tokenized raw text in Wikitext-2 dataset [Merity et al., 2017]. The query, key and value weight matrices in initialized or pretrained models transform input $X$ into $Q, K, V$ of different distributions. We compare the results with different sequence lengths and different numbers of features used in attention approximation methods. We set the number of features in the range of $2^4$ to $2^8$. More features usually require more computation resources.

Figure 1 shows the performance of the modified Nyström method on approximation error with regards to the number of features. We conclude that for Skyformer the approximation is significantly better with the increased number of features, while for other methods the gain is not obvious. The good performance of the modified Nyström method also validates our previous claim that the Nyström method is currently one of the most powerful methods in large-scale kernel machines acceleration.

**Remark.** Although in a single step the modified Nyström method in Section 4.2 can give low approximation error, we do not recommend directly applying it to the original self-attention. With some exploratory experiments on classification tasks, we find the variant suffers a more severe gradient explosion issue than usual transformers. We speculate that it is because the matrix $\boldsymbol{S}^T\bar{\boldsymbol{A}}\mathbf{S}$ (in the middle of Equation (6)) inherits the high condition number of the original attention score matrices $\boldsymbol{A}$, while the derivative of matrix inverse $\left(\left(\boldsymbol{A}^{-1}\right)' = -\boldsymbol{A}^{-1}\boldsymbol{A}'\boldsymbol{A}^{-1}\right)$ further amplifies the condition number during backpropagation.

## 5 Experimental Results

**Tasks and Datasets.** We evaluate the proposed methods on five classification tasks on LRA benchmark [Tay et al., 2020b], which focuses on model quality under long-context scenarios: ListOps [Nangia and Bowman, 2018], Text Classification on IMDb review dataset [Maas et al., 2011], Document Retrieval on AAN dataset [Radev et al., 2013], Pathfinder [Linsley et al., 2018], and Image Classification on CIFAR-10 [Krizhevsky et al., 2009]. The LRA benchmark covers diverse long-sequence tasks in sequence length, task difficulty, and inspected model abilities. For example, ListOps and Pathfinder evaluate the abilities to capture the long-range hierarchical dependency and spatial dependency, respectively, which poses challenges for sparse attention pattern based methods. We

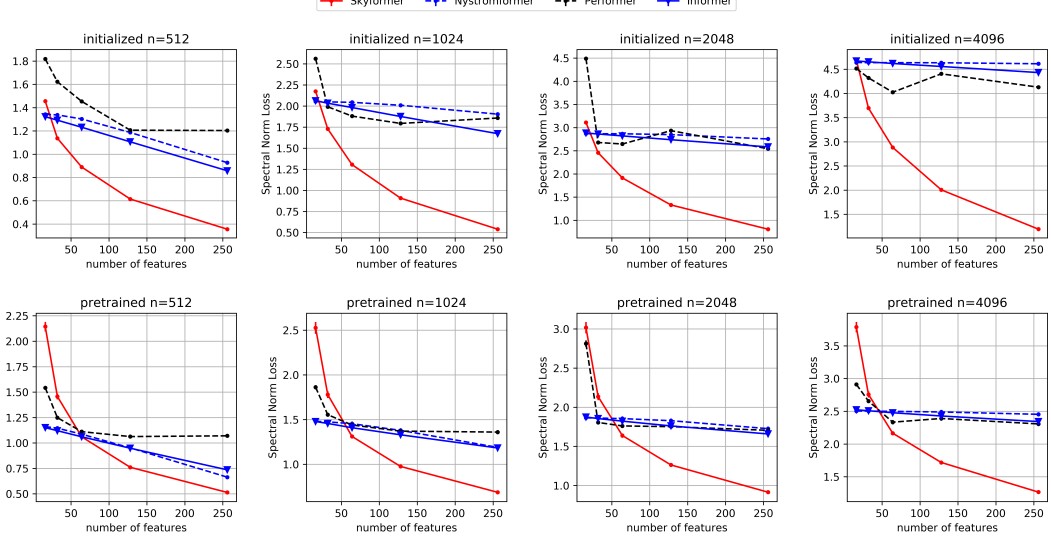

Figure 1: Spectral norm results with different sequence lengths under different $W_Q, W_K, W_V$ settings, either from initialized or pretrained BERT models. All methods are approximating the original self-attention output. Y axis: Lower spectral loss means better approximation. X axis: Higher $d$ (number of features) means visiting more elements in the original matrix and bringing more computation costs. The label "Skyformer" here means that we use the algorithm behind Skyformer, mainly Eq. (5), to approximate the raw attention score matrix $A$ in self-attention. In this experiment, "Skyformer" also needs to first approximate $A$, and then approximate $D$, as Performer does.

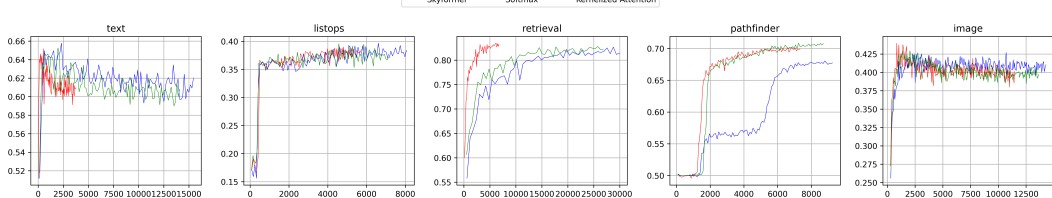

Figure 2: Validation accuracy changes with respect to training time for $50k$ steps. X axis: training time (s). Y axis: classification accuracy.

Table 1: Classification accuracy (%) on LRA benchmark.

| Model | Text | ListOps | Retrieval | Pathfinder | Image | AVG. |
|---|---|---|---|---|---|---|
| Self-Attention | 61.95 | 38.37 | 80.69 | 65.26 | 40.57 | 57.37 |
| Kernelized Attention | 60.22 | 38.78 | 81.77 | 70.73 | 41.29 | 58.56 |
| Nystromformer | 64.83 | 38.51 | 80.52 | 69.48 | 41.30 | 58.93 |
| Linformer | 58.93 | 37.45 | 78.19 | 60.93 | 37.96 | 54.69 |
| Informer | 62.64 | 32.53 | 77.57 | 57.83 | 38.10 | 53.73 |
| Performer | 64.19 | 38.02 | 80.04 | 66.30 | 41.43 | 58.00 |
| Reformer | 62.93 | 37.68 | 78.99 | 66.49 | 48.87 | 58.99 |
| BigBird | 63.86 | 39.25 | 80.28 | 68.72 | 43.16 | 59.05 |
| **Skyformer** | 64.70 | 38.69 | 82.06 | 70.73 | 40.77 | **59.39** |

Table 2: Running time (hour) and peak memory usage (GB). **TC**: Text Classification. **LO**: ListOps. **RE**: Retrieval. **PF**: Pathfinder. **IC**: Image Classification. **KA**: Kernelized Attention.

| Model | Time (h) | | | | | Memory (GB) | | | | |
| --- | --- | --- | --- | --- | --- | --- | --- | --- | --- | --- |
| | TC | LO | RE | PF | IC | TC | LO | RE | PF | IC |
| Self-Attention | 4.30 | 2.24 | 8.33 | 2.57 | 4.22 | 10.37 | 5.37 | 10.77 | 5.74 | 11.47 |
| KA | 3.91 | 1.99 | 7.46 | 2.42 | 4.05 | 5.73 | 5.94 | 10.46 | 6.38 | 6.38 |
| Nystromformer | 0.71 | 0.71 | 1.29 | 1.49 | 2.70 | 1.21 | 1.37 | 2.39 | 3.35 | 6.71 |
| Linformer | 0.65 | 0.60 | 1.13 | 1.09 | 2.19 | 0.99 | 0.99 | 1.89 | 1.97 | 3.94 |
| Informer | 1.60 | 1.19 | 2.91 | 2.39 | 3.90 | 5.12 | 4.85 | 5.77 | 4.75 | 9.51 |
| Performer | 0.77 | 0.73 | 1.41 | 1.40 | 2.55 | 1.09 | 1.09 | 2.16 | 2.20 | 4.39 |
| Reformer | 0.94 | 0.85 | 1.73 | 1.70 | 3.08 | 1.61 | 1.61 | 2.98 | 3.21 | 6.42 |
| BigBird | 2.00 | 1.88 | 3.81 | 3.39 | 6.53 | 2.83 | 2.71 | 4.97 | 4.97 | 9.95 |
| Skyformer | 1.02 | 1.29 | 1.86 | 2.03 | 3.40 | 1.59 | 1.75 | 3.15 | 4.13 | 8.26 |

report the classification accuracy on the test set, training time, and peak memory usage during training for each task.

**Baselines.** Aside from the vanilla quadratic self-attention, we compare with Big Bird [Zaheer et al., 2020], Performer [Choromanski et al., 2020], Linformer [Wang et al., 2020], Nyströmformer [Xiong et al., 2021], Informer [Zhou et al., 2020], and Reformer [Kitaev et al., 2020]. Most methods are approximating the vanilla full attention for efficiency and thus are not expected to have better performance. As it is not realistic to exhaustively fine-tune all models and search for the best performance under limited computation resources, we instead only replace the self-attention module with the various attention methods and keep other experimental settings the same for fair comparisons.

**Implementation Details.** We conduct each experiment on one Tesla V100 SXM2 16GB. We use the LRA evaluation benchmark reimplemented in PyTorch by Xiong et al. [2021]. We use a 2-layer transformer model with $64$ embedding dimension, $128$ hidden dimension, $2$ attention heads, and mean pooling for classification. Batch size is selected conditioned on the memory requirements of the standard self-attention method, which leads to $16$ for Text Classification, $32$ for ListOps, $16$ for Document Retrieval, $128$ for Pathfinder, and $256$ for Image Classification. Learning rate is set to $1e-4$ for Text Classification, ListOps, and Image Classification, and $2e-4$ for Retrieval and Pathfinder. Each model on each task is trained for $50k$ steps, during which the best checkpoint with the highest accuracy on the development set will be saved for evaluation. For comparable computation complexity, we control the number of features to be $128$ used in all methods (except Big Bird), under which setting the models will visit $128 \cdot n$ elements in the attention matrix. For numerical consistency, all experiment results are averaged across three runs with different random seeds.

We do not follow all settings in [Xiong et al., 2021] due to the hardware limitation. The compromises, such as approximation dimension and gradient accumulations steps, might bring performance differences comparing to results reported in [Xiong et al., 2021]. The training instability problem also helps explain the performance gap.

**Results.** The training process of the standard softmax-based method is unstable as observed in Figure 2: it takes more steps to reach the stationary distribution of its long-time limit, and it is more easily getting stuck in a local minimum. Runs with different random seeds may bring divergent performances, and probably leads to lower averaged scores. We have also tried directly approximating the self-attention method with the Nyström method and observed numerical instability during training.

Replacing the softmax structure with Gaussian kernel somehow alleviates this instability problem with boosted performance as shown in Table 1. However, the time and space requirement of Kernelized Attention is not significantly improved compared to the original version, which serves as the motivation to approximate Kernelized Self-Attention with Nyström method.

Though not necessarily the fastest, our proposed Skyformer can efficiently converge to the long-time limit with comparable general performance in classification accuracy (Table 1) and resource consumption (Table 2). The advantages over the standard self-attention are significant with consistently less training time and generally better performance. For example, Skyformer brings nearly 4

times speed-up on text classification and document retrieval while with 2.75% and 1.37% accuracy improvement over the standard self-attention.

**Limitations.** The applications of Skyformer might be limited to long sequence tasks because for small sequence length $n$ the statistical dimension $d_stat$ might be close to $n$.

To make the claim above clear, we first reiterate that the efficiency of Skyformer is related to $d_stat$. As implied by Theorem 2, the intrinsic difficulty of approximating a raw attention score matrix is concluded as $d_{stat}$, which corresponds to the effective rank of matrix $\bar{C}$. The complexity of Skyformer depends on the sub-sample size $d$ (the size of the sub-sampling matrix $S$). A large $d_{stat}$ leads to a large $d$, and an inefficient application of the Nyström method.

The classical theory for statistical dimension only guarantees that $d_{stat}$ is small (compared to $n$) when $n$ is large enough, and it is possible the statistical dimension associated with a short sequence might be even close to the sequence $n$. Therefore a large $n$ serves as a condition to make the method work. Figure 1 empirically shows that our method performs better with larger $n$'s.

# 6    Conclusions and future work

Motivated by the connection between kernel methods and self-attention, we introduce Kernelized Attention, which replaces the softmax structure in self-attention with a Gaussian kernel. We also propose Skyformer, which adapts the Nyström method to Kernelized Attention to improve its efficiency. We expect the new model can enjoy more stable training while inheriting the strong performance from self-attention. Extensive experiments verify our intuitions and show that both Kernelized Attention and its Nyström approximation variant have comparable accuracy to the original Transformer on the LRA benchmark.

Direct development of this work is the incorporation of further computation tricks in kernel methods, such as the local and global approximation for gram matrix [Snelson and Ghahramani, 2007] and the importance sampling in Nyström methods [Musco and Musco, 2017, Chen and Yang, 2021b,a]. Other related questions include the choice of the kernel other than the Gaussian kernel in our kernelized attention model. It is expected that for different tasks there will be specific kernels more proper than the original self-attention. The results in this work also shed new light on the design of the attention mechanism, which may benefit board downstream NLP tasks.

## Acknowledgments and Disclosure of Funding

This research is based upon work in part supported by the Office of the Director of National Intelligence (ODNI), Intelligence Advanced Research Projects Activity (IARPA), via contract No. FA8650-17-C-9116, and U.S. DARPA KAIROS Program No. FA8750-19-2-1004. This work is also in part supported by NSF grant DMS-1810831. The views and conclusions contained herein are those of the authors and should not be interpreted as necessarily representing the official policies, either expressed or implied, of DARPA, ODNI, IARPA, or the U.S. Government. The U.S. Government is authorized to reproduce and distribute reprints for governmental purposes notwithstanding any copyright annotation therein.

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
