# Appendix: Remodel Self-Attention with Gaussian Kernel and Nyström Method

**Yifan Chen,**[*] **Qi Zeng,**[*] **Heng Ji, Yun Yang**
University of Illinois Urbana-Champaign
{yifanc10, qizeng2, hengji, yy84}@illinois.edu

## A    Validation loss

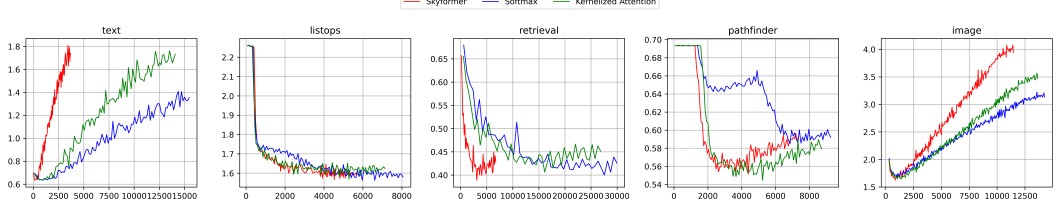

Figure 1:   Validation loss changes for $50k$ steps. X-axis: Training time (second). Y-axis: Cross Entropy Loss on validation set.

Figure 1 shows the validation loss changes with respect to training time for 50k steps as supplementary results for the experiments in Section 5. In general, Skyformer converges faster and finishes 50k steps earlier than vanilla Attention and Kernelized Attention over all tasks. We further remark that on Text Classification, all models quickly fall into over-fitting, and thus the validation losses rise quickly. On Pathfinder, due to the difficulty of training, in the trial shown in the figure vanilla Attention fails to reach the best long-time limit under a certain setting.

## B    Singular value decay rate

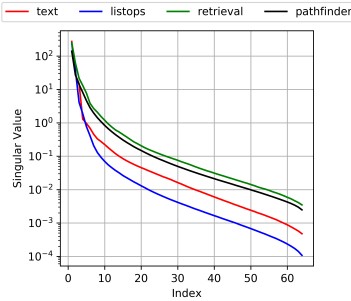

Figure 2:   Singular value distribution of attention output.

Figure 2 shows the singular value distribution of attention output from the second layer of a trained vanilla transformer. Results are averaged across one random batch from the test set in each LRA task.

---

[*]   Equal contribution.

35th Conference on Neural Information Processing Systems (NeurIPS 2021).

The singular values decay fast and thus justify the low-rank approximation, as analyzed by Wang et al. [2020], Dong et al. [2021]. We propose to measure the task difficulty with the singular value decay rate in attention output, as higher intrinsic task difficulty forces the model to output a matrix with more large singular values. Such matrices are considered more informative since they are harder to approximate, requiring more ranks even in the truncated SVD approximation. With the observation in Figure 2, we conclude that the singular values in Document Retrieval and Pathfinder tasks decay slower, and those two tasks are more difficult than Text Classification and ListOps.

## C   Useful facts

This section introduces some useful facts, which are key in the proof in the next section. To start with, we provide a matrix concentration inequality as follows.

**Lemma C.1** (Matrix Bernstein Inequality [Tropp, 2012]). *Consider a finite sequence $\{\boldsymbol{X}_k\}$ of independent, random, self-adjoint matrices with dimension $n$. Assume that each random matrix satisfies*

$$\mathbb{E}\,\boldsymbol{X}_k = \boldsymbol{0} \quad and \quad \|\boldsymbol{X}_k\| \leq R \quad almost\ surely.$$

*Then, for all $t \geq 0$,*

$$\mathbb{P}\left\{\|\sum_k \boldsymbol{X}_k\| \geq t\right\} \leq 2n \cdot \exp\left(\frac{-t^2/2}{\sigma^2 + Rt/3}\right) \quad where \quad \sigma^2 \geq \left\|\sum_k \mathbb{E}\left(\boldsymbol{X}_k^2\right)\right\|.$$

For a certain $n$-by-$n$ orthogonal matrix $\boldsymbol{H}$ ($\boldsymbol{H}\boldsymbol{H}^T$ is a diagonal matrix) and an $n$-by-$d$ uniform sub-sampling matrix $\boldsymbol{S}$ (as defined in Definition 1 in the main paper), we denote the sketching matrix $\boldsymbol{\Pi} := \sqrt{n}\boldsymbol{S}$. We aim to show $\boldsymbol{H}\boldsymbol{\Pi}\boldsymbol{\Pi}^T\boldsymbol{H}^T$ can satisfy $(\frac{1}{2}, \delta)$-MA property for $\boldsymbol{H}\boldsymbol{H}^T$ by the following lemma.

**Lemma C.2.** *Denote the stable rank $s := \frac{\|\boldsymbol{H}\|_F^2}{\|\boldsymbol{H}\|^2} \geq 1$, and a constant $\delta < 1/2$. Suppose there exists a constant $\beta \in (0, 1]$ such that $\beta \leq \frac{\|\boldsymbol{H}\|_F^2}{n\|\boldsymbol{H}^{(i)}\|^2}, \forall i = 1, \ldots, n$, where $\boldsymbol{H}^{(i)}$ is the $i$-th column of $\boldsymbol{H}$. There exists a constant $C_0$ that if*

$$d \geq C_0 \frac{s}{\beta} \log \frac{n}{\delta},$$

*then $\boldsymbol{H}\boldsymbol{\Pi}\boldsymbol{\Pi}^T\boldsymbol{H}^T$ satisfies $(\frac{1}{2}, \delta)$-MA property for $\boldsymbol{H}\boldsymbol{H}^T$.*

*Proof.* The main idea is to utilize Lemma C.1 by setting $t = \frac{1}{2}\|\boldsymbol{H}\boldsymbol{H}^T\| = \frac{1}{2}\|\boldsymbol{H}\|^2$. Specifically, we denote the matrices

$$\boldsymbol{X}_k = \boldsymbol{H}\boldsymbol{\Pi}^{(i)}\left(\boldsymbol{\Pi}^{(i)}\right)^T \boldsymbol{H}^T - \frac{1}{d}\boldsymbol{H}\boldsymbol{H}^T, \quad \text{so that}$$

$$\sum_k \boldsymbol{X}_k = \boldsymbol{H}\boldsymbol{\Pi}\boldsymbol{\Pi}^T\boldsymbol{H}^T - \boldsymbol{H}\boldsymbol{H}^T.$$

We still need two steps to give control of $R$ and $\sigma^2$. For $R$, we have

$$\|\boldsymbol{X}_k\| = \left\|\frac{1}{d}\sum_{i=1}^n (nz_{ki} - 1)\boldsymbol{H}^{(i)}\left(\boldsymbol{H}^{(i)}\right)^T\right\| \leq \frac{1}{d}\max\left\{\max_i (n-1)\|\boldsymbol{H}^{(i)}\|^2, \|\boldsymbol{H}\|^2\right\}$$

$$\leq \frac{1}{d}n\max_i\|\boldsymbol{H}^{(i)}\|^2,$$

where $\{z_{ki}\}_{i=1}^n$ are the indicators of whether the $i$-th column is chosen. The first inequality of the preceding display holds due to the fact that $\boldsymbol{H}$ is an orthogonal matrix. Using the condition $n \leq \frac{\|\boldsymbol{H}\|_F^2}{\beta\|\boldsymbol{H}^{(i)}\|^2}, \forall i = 1, \ldots, n$, we further have

$$\|\boldsymbol{X}_k\| \leq \frac{\|\boldsymbol{H}\|_F^2}{d\beta},$$

and we thus set $R := \frac{\|\boldsymbol{H}\|_F^2}{d\beta}$. On the other hand,

$$\mathbb{E}\,\boldsymbol{X}_k^2 = \frac{1}{d^2}\sum_{i=1}^n \mathbb{E}\left((nz_{ki}-1)^2\right)\|\boldsymbol{H}^{(i)}\|^2 \boldsymbol{H}^{(i)}\left(\boldsymbol{H}^{(i)}\right)^T = \frac{1}{d^2}\sum_{i=1}^n (n-1)\|\boldsymbol{H}^{(i)}\|^2 \boldsymbol{H}^{(i)}\left(\boldsymbol{H}^{(i)}\right)^T.$$

Again using the condition that $n\|\boldsymbol{H}^{(i)}\|^2 \le \frac{\|\boldsymbol{H}\|_F^2}{\beta}, \forall i = 1,\ldots,n$, we reach

$$\left\|\mathbb{E}\sum_{k=1}^d \boldsymbol{X}_k^2\right\| \le \frac{1}{d}\frac{\|\boldsymbol{H}\|_F^2}{\beta}\left\|\boldsymbol{H}\boldsymbol{H}^T\right\| = \frac{\|\boldsymbol{H}\|_F^2}{d\beta}\|\boldsymbol{H}\|^2,$$

and set $\sigma^2 := \frac{\|\boldsymbol{H}\|_F^2}{d\beta}\|\boldsymbol{H}\|^2$.

Finally we plug $R$ and $\sigma^2$ into Lemma C.1 and obtain:

$$\mathbb{P}\left\{\|\boldsymbol{H}\boldsymbol{\Pi}\boldsymbol{\Pi}^T\boldsymbol{H}^T - \boldsymbol{H}\boldsymbol{H}^T\| \ge \frac{1}{2}\|\boldsymbol{H}\|^2\right\} \le 2n \cdot \exp\left(\frac{-\|\boldsymbol{H}\|^4/8}{\frac{s\|\boldsymbol{H}\|^4}{d\beta} + \frac{s\|\boldsymbol{H}\|^4}{6d\beta}}\right).$$

To ensure the right-hand-side is smaller than $\delta$, we just need

$$d \ge \frac{28}{3}\frac{s}{\beta}\log\frac{2n}{\delta},$$

which validates the lemma. $\diamondsuit$

## D  Proof of Theorem 2 in the main paper

*Proof.* The conclusion in the lemma can be divided into two parts, that $\tilde{\bar{C}} \preccurlyeq \bar{C}$ and $\bar{C} \preccurlyeq \tilde{\bar{C}} + \lambda\boldsymbol{I}$. To prove them we first introduce some notations and auxiliary results. Since $\bar{C}$ is PSD, there exists a matrix $\boldsymbol{B}$ satisfying $\boldsymbol{B}\boldsymbol{B}^T = \bar{C}$. We further denote $\boldsymbol{B}$'s SVD decomposition as $\boldsymbol{B} = \boldsymbol{U}\boldsymbol{\Sigma}^{\frac{1}{2}}\boldsymbol{V}^T$ ($\bar{C} = \boldsymbol{U}\boldsymbol{\Sigma}\boldsymbol{U}^T$), where both $\boldsymbol{U}$ and $\boldsymbol{V}$ are $2n$-by-$2n$ orthonormal matrices. (In this section we slightly abuse the notation that $\boldsymbol{V}$ represents the matrix of right-singular vectors, instead of the value matrix in self-attention.) Define $\bar{\boldsymbol{\Sigma}} := \boldsymbol{\Sigma} + \lambda\boldsymbol{I}, \boldsymbol{\Psi} := \boldsymbol{U}\boldsymbol{\Sigma}^{\frac{1}{2}}\bar{\boldsymbol{\Sigma}}^{-\frac{1}{2}}$, which implies $\bar{C}(\bar{C}+\lambda\boldsymbol{I})^{-1} = \boldsymbol{\Psi}\boldsymbol{\Psi}^T$. Also following the notations in the last section, we define the $2n$-by-$d$ matrix $\boldsymbol{\Pi} := \sqrt{2n}\boldsymbol{S}$ 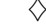 With those notations, $\tilde{\bar{C}}$ can be rewritten as $\boldsymbol{B}\boldsymbol{B}^T\boldsymbol{\Pi}(\boldsymbol{\Pi}^T\boldsymbol{B}\boldsymbol{B}^T\boldsymbol{\Pi})^\dagger\boldsymbol{\Pi}\boldsymbol{B}\boldsymbol{B}^T = \boldsymbol{B}\boldsymbol{P}_{\boldsymbol{\Pi}}\boldsymbol{B}^T$, where $\boldsymbol{P}_{\boldsymbol{\Pi}}$ is the orthogonal projection matrix for the column space of $\boldsymbol{B}^T\boldsymbol{\Pi}$. It is easy to check that $\bar{C} - \tilde{\bar{C}} = \boldsymbol{B}(\boldsymbol{I} - \boldsymbol{P}_{\boldsymbol{\Pi}})\boldsymbol{B}^T$. Since $\boldsymbol{I} - \boldsymbol{P}_{\boldsymbol{\Pi}}$ is an orthogonal projection matrix (which is PSD), we have $\bar{C} - \tilde{\bar{C}} \succcurlyeq \boldsymbol{0}$, which proves the first conclusion that $\tilde{\bar{C}} \preccurlyeq \bar{C}$.

For the second conclusion, we utilize the following important identity:

$$\boldsymbol{B}^T\boldsymbol{\Pi}\boldsymbol{\Pi}^T\boldsymbol{B} - \boldsymbol{B}^T\boldsymbol{B} = \boldsymbol{V}\bar{\boldsymbol{\Sigma}}^{\frac{1}{2}}\left(\bar{\boldsymbol{\Sigma}}^{-\frac{1}{2}}\boldsymbol{V}^T(\boldsymbol{B}^T\boldsymbol{\Pi}\boldsymbol{\Pi}^T\boldsymbol{B} - \boldsymbol{B}^T\boldsymbol{B})\boldsymbol{V}\bar{\boldsymbol{\Sigma}}^{-\frac{1}{2}}\right)\bar{\boldsymbol{\Sigma}}^{\frac{1}{2}}\boldsymbol{V}^T$$
$$= \boldsymbol{V}\bar{\boldsymbol{\Sigma}}^{\frac{1}{2}}\left(\boldsymbol{\Psi}^T\boldsymbol{\Pi}\boldsymbol{\Pi}^T\boldsymbol{\Psi} - \boldsymbol{\Psi}^T\boldsymbol{\Psi}\right)\bar{\boldsymbol{\Sigma}}^{\frac{1}{2}}\boldsymbol{V}^T.$$

For $\boldsymbol{\Psi}$, we have that its squared Frobenius norm $\|\boldsymbol{\Psi}\|_F^2 = d_{stat}$, and $\|\boldsymbol{\Psi}\|^2 = \|\bar{C}(\tilde{\bar{C}}+\lambda\boldsymbol{I})^{-1}\| \ge 1/2$, indicating that $\boldsymbol{\Psi}$'s stable rank $s = \|\boldsymbol{\Psi}\|_F^2 / \|\boldsymbol{\Psi}\|^2$ is at most $2d_{stat}$.

Taking $\varepsilon = \frac{1}{2}$ and applying Lemma C.2, we can conclude that with the conditions on $d$ in the theorem, $\boldsymbol{\Psi}^T\boldsymbol{\Pi}\boldsymbol{\Pi}^T\boldsymbol{\Psi}$ satisfies $(\frac{1}{2},\delta)$-MA property for $\boldsymbol{\Psi}^T\boldsymbol{\Psi}$. Therefore it holds with probability $1 - \delta$ that,

$$\|\boldsymbol{\Psi}^T\boldsymbol{\Pi}\boldsymbol{\Pi}^T\boldsymbol{\Psi} - \boldsymbol{\Psi}^T\boldsymbol{\Psi}\| \le \frac{1}{2}\|\boldsymbol{\Psi}\|^2 \le \frac{1}{2}.$$

From identity $\boldsymbol{V}\bar{\boldsymbol{\Sigma}}^{\frac{1}{2}}\bar{\boldsymbol{\Sigma}}^{\frac{1}{2}}\boldsymbol{V}^T = \boldsymbol{B}^T\boldsymbol{B} + \frac{\lambda}{2}\boldsymbol{I}$, we obtain

$$\frac{1}{2}\boldsymbol{B}^T\boldsymbol{B} - \frac{\lambda}{2}\boldsymbol{I} \preccurlyeq \boldsymbol{B}^T\boldsymbol{\Pi}\boldsymbol{\Pi}^T\boldsymbol{B} \preccurlyeq \frac{3}{2}\boldsymbol{B}^T\boldsymbol{B} + \frac{\lambda}{2}\boldsymbol{I}, \tag{1}$$

which implies

$$\boldsymbol{B}^T\boldsymbol{B} \preccurlyeq 2\boldsymbol{B}^T\boldsymbol{\Pi}\boldsymbol{\Pi}^T\boldsymbol{B} + \lambda\boldsymbol{I}. \tag{2}$$

Finally, we multiply two sides of Eq. (2) by $(\boldsymbol{I} - \boldsymbol{P_\Pi})$ to obtain

$$(\boldsymbol{I} - \boldsymbol{P_\Pi})\boldsymbol{B}^T\boldsymbol{B}(\boldsymbol{I} - \boldsymbol{P_\Pi}) \preccurlyeq 2 \cdot 0 + \lambda(\boldsymbol{I} - \boldsymbol{P_\Pi}) \preccurlyeq \lambda\boldsymbol{I},$$

where the second inequality is due to the fact that $(\boldsymbol{I} - \boldsymbol{P_\Pi})$ is an orthogonal projection matrix. The equation above implies $\|(\boldsymbol{I} - \boldsymbol{P_\Pi})\boldsymbol{B}^T\boldsymbol{B}(\boldsymbol{I} - \boldsymbol{P_\Pi})\| = \|\boldsymbol{B}(\boldsymbol{I} - \boldsymbol{P_\Pi})\boldsymbol{B}^T\| \leq \lambda$, which completes the proof for the second conclusion $\bar{\boldsymbol{C}} \preccurlyeq \tilde{\bar{\boldsymbol{C}}} + \lambda\boldsymbol{I}$.

Based on the conclusion above, the last implication is direct:

$$\|\tilde{\boldsymbol{C}} - \boldsymbol{C}\| = \left\|(\boldsymbol{I}, \boldsymbol{0})\left(\tilde{\bar{\boldsymbol{C}}} - \bar{\boldsymbol{C}}\right)(\boldsymbol{0}, \boldsymbol{I})^T\right\| \leq \|(\boldsymbol{I}, \boldsymbol{0})\|\left\|\left(\tilde{\bar{\boldsymbol{C}}} - \bar{\boldsymbol{C}}\right)\right\|\|(\boldsymbol{0}, \boldsymbol{I})^T\| \leq \lambda = \varepsilon\|\boldsymbol{C}\|,$$

which completes the proof. $\diamond$

## E  Proof of Lemma 3 in the main paper

*Proof.* As $\bar{\boldsymbol{C}}$ is constructed based on a PSD kernel, $\bar{\boldsymbol{C}}$ is also PSD. Consequently $\boldsymbol{M} = \boldsymbol{S}^T\bar{\boldsymbol{C}}\boldsymbol{S}$ is PSD, and $\boldsymbol{D}_M^{-1/2}(\boldsymbol{M} + \gamma\boldsymbol{I})\boldsymbol{D}_M^{-1/2}$ is positive definite, with all eigenvalues positive. To prove the claim in the lemma we only need to show the eigenvalues of $\boldsymbol{D}_M^{-1/2}(\boldsymbol{M} + \gamma\boldsymbol{I})\boldsymbol{D}_M^{-1/2}$ are bounded from above by 1. It is equivalent to prove that $\boldsymbol{I} - \boldsymbol{D}_M^{-1/2}(\boldsymbol{M} + \gamma\boldsymbol{I})\boldsymbol{D}_M^{-1/2}$ is PSD, which can be induced by another statement that $\boldsymbol{L} := \boldsymbol{D}_M - (\boldsymbol{M} + \gamma\boldsymbol{I})$ is PSD.

The proof of the statement above is similar to the proof of the well-known conclusion that graph Laplacian matrix is PSD. For simplicity we denote $\boldsymbol{W} := \boldsymbol{M} + \gamma\boldsymbol{I}$, and given any vector $\boldsymbol{x} \in \mathbb{R}^d$ we have

$$\boldsymbol{x}^T\boldsymbol{L}\boldsymbol{x} = \boldsymbol{x}^T\boldsymbol{D}_M\boldsymbol{x} - \boldsymbol{x}^T\boldsymbol{W}\boldsymbol{x} = \sum_{i=1}^d (\boldsymbol{D}_M)_{ii}\boldsymbol{x}_i^2 - \sum_{i,j=1}^d \boldsymbol{W}_{ij}\boldsymbol{x}_i\boldsymbol{x}_j$$

$$= \frac{1}{2}\left(\sum_{i=1}^d (\boldsymbol{D}_M)_{ii}\boldsymbol{x}_i^2 - 2\sum_{i,j=1}^d \boldsymbol{W}_{ij}\boldsymbol{x}_i\boldsymbol{x}_j + \sum_{j=1}^d (\boldsymbol{D}_M)_{jj}\boldsymbol{x}_j^2\right)$$

$$= \frac{1}{2}\sum_{i,j=1}^d \boldsymbol{W}_{ij}(\boldsymbol{x}_i - \boldsymbol{x}_j)^2 \geq 0,$$

where the last equation holds due to the fact that $(\boldsymbol{D}_M)_{ii} = \sum_{j=1}^d \boldsymbol{W}_{ij}$.

Combining the pieces above we can conclude that $\|\boldsymbol{I} - \boldsymbol{D}_M^{-1/2}(\boldsymbol{M} + \gamma\boldsymbol{I})\boldsymbol{D}_M^{-1/2}\| < 1$. $\diamond$

## F  Additional discussions about the stability in model training

For our argument about stability, we mainly refer to the paper [Liu et al., 2020], which identifies that the amplification of small parameter perturbations in the self-attention module is the root cause of training instability. We take kernelized attention as mitigation since it contains an automatic normalization. We have empirically used Figure 2 and Figure 1 in Appendix A to support our claim.

For further analysis we conduct a toy experiment adapting from Figure 4 in the aforementioned paper [Liu et al., 2020]. We aim to show that in kernelized attention (and Skyformer) the output changes $f(x, W^*) - f(x, W)$ for parameter changes $W^* - W$ is smaller than in self-attention (and its approximation Nyströmformer). This concept involved is somewhat similar to condition number and below we will formalize it as "instability score".

We show a table of the averaged ratios between the instability scores of kernelized attention (we also add Skyformer and Nyströmformer for reference) and self-attention to conclude our statement about stability. A ratio smaller than 1 means higher stability compared to self-attention. We follow all the settings in Table 1 in the main paper except here we only update the model for 20 steps (we limit the number of steps as suggested by Liu et al. [2020] to make the results of the same step comparable among different models). In step $i$ for each model we compute the instability score

Table 1: Ratios of instability score on LRA benchmark.

| Model | Text | ListOps | Retrieval | Pathfinder | Image |
|---|---|---|---|---|---|
| Nyströmformer | 1.03 | 1.01 | 0.97 | 0.99 | 1.02 |
| Kernelized Attention | 0.83 | 0.77 | 0.64 | 0.74 | 0.62 |
| Skyformer | 0.81 | 0.79 | 0.64 | 0.79 | 0.65 |

$\tau_i = \frac{\|f(x_i, W_i) - f(x_i, W_{i-1})\|_F^2}{\|W_i - W_{i-1}\|_F^2}, i = 1, \cdots, 20$, where $f()$ gives the embedding after two layers, $x_i$ is the $i$-th input sequence batch, $W_0$ represents the initial parameters, and $W_i$ represents the parameters after step $i$. In each step we compute the ratio of a certain method's $\tau_i$ to the $\tau_i$ of self-attention, and finally average the 20 ratios in Table 1 in the appendix.

As we can observe, both kernelized attention and Skyformer consistently have a lower instability score than self-attention, while the instability score of Nyströmformer, an approximation to self-attention, fluctuates around 1 in all the tasks. The results support our claim that the proposed kernelized attention can improve stability.