# OpenReview forum: "Skyformer: Remodel Self-Attention with Gaussian Kernel and Nystr\"om Method"
_NeurIPS.cc/2021/Conference — NeurIPS 2021 Poster_

### Official Review · Reviewer_4vy6 · 2021-07-17

**Rating:** 7
**Confidence:** 5

**Summary:**

The authors reformulate soft max structure of the self-attention mechanism in Transformers in terms of a Gaussian kernel evaluation. They claim that such a model is empirically stable during training, while allowing the use of various Kernel approximation strategies to improve the quadratic computational complexity.

In this work, approximation strategy used by the authors is the popular Nyström approximation. However, since the kernel matrix formed between the query and the key matrices is not symmetric (hence not PSD), the Nyström approximation is not directly applicable. The authors propose to form a PSD kernel matrix by first evaluating the Gaussian kernel on the row concatenated  query and key matrices which is PSD, performing the Nyström approximation and then choosing the appropriate block in the resulting matrix.

By adapting an existing theorem [Musco, Musco 17],  the authors claim that such a low-rank approximation of the kernelized attention score is bounded in terms of the relative spectral norm. This is claimed to be due to the observed fast decay of the eigenvalues in an empirical kernel matrix.

The matrix inversion required for the Nyström approximation, however is a potential issue in terms of numerical stability on a GPU. To circumvent the problem, instead of a standard conjugate gradient method, they use a Schulz type iteration which only uses matrix products, without any floating point divisions.

Empirical evaluation shows marginal improvement on most tasks on the LRA benchmark, as compared to the naive Nyström approximation [Xiong et al 2021]. However, this is at the cost of relatively larger time and memory usage.

**Limitations And Societal Impact:**

The authors did not point out or address any potential negative societal impact of their work. Being essentially a theoretical improvement over previous work, there is limited potential for excess social impact as compared to previous work in this field.

**Main Review:**

Originality:
The main contribution of this work seems to be the proposed fix to the theoretical issue of Nyström approximation in [Xiong et al 2021]. The rest of the methods used here are priorly known techniques, though presented in a cogent manner. They cite most important of the previous works, however at times their contributions are not clearly delineated from existing methods. For example, the Kernelized attention model (line 54). Reference to [Chormanski et al 2020] for Lemma 1 is also missing.

Quality:
The submission is technically sound and correctly applies the Nyström approximation to the kernelized attention mechanism. The claim of good approximation in spectral norm is supported by a stated theorem as well as empirical evaluation. However, the use of Schulz type iteration for inverse may have the unintended consequence of "zero fill in", i.e. the approximate inverse converges to a dense matrix even when the kernel matrix was sparse.

Clarity:
The paper is well written and clearly organised. Claims of speed-up though (line 313) might be considered misleading as they direct competitor is the method in [Xiong et al 2021]. The experimental setup is fairly well described with the code being adapted from [Xiong et al 2021].

Significance:
The paper is good contribution towards the development of theoretically sound approximations and reformulations of the self-attention mechanism.

**Time Spent Reviewing:**

3

---

> ### Author Response · Authors · 2021-08-10
> **Author Response of Paper512**
>
> We gratefully acknowledge the in-depth reviews and the helpful comments.
>
> We will incorporate your advice into the next version by carefully adding more discussion on the related work, the potential numerical issue while using a sparse kernel matrix, and the direct comparison with Nyströmformer regarding training efficiency.

---

### Official Review · Reviewer_7p5f · 2021-07-17

**Rating:** 6
**Confidence:** 3

**Summary:**

The paper proposes a new form of self-attention for use in transformer models using Gaussian Kernels. They show that the new attention method is comparable in accuracy to traditional softmax self-attention. They then go on to show that kernel based attention can be approximated by taking a Nystrom sketch of the kernel matrix. They give rigorous bounds on the spectral norm of the approximation (assuming that the kernel matrix has a high condition number).

**Limitations And Societal Impact:**

Don't see any potential negative societal impact from reducing resources required for training and inferencing

**Main Review:**

Pros
* Proposal of a new attention method (kernel based instead of soft-max)
* Rigorous proof and a proper sketch of the scenarios where one can use Nystrom method. Mathematically founded approximation of the kernel matrix unlike earlier approach (Nystromformer)
* Mostly clear and easy to follow

Cons
* Work seems incremental (Nystromformer is very similar). But the nice thing is the authors do a good job of giving proofs motivating their algorithm

Nit
* Page 1: Introduction: O(n^2) not O(n)

**Time Spent Reviewing:**

4

---

> ### Author Response · Authors · 2021-08-10
> **Author Response of Paper512**
>
> We appreciate your useful comments. We argue that Nyströmformer and our model seem to be alike only because both are based on the Nyström method. By highlighting the contributions of our work we hope to address your concern.
>
> Compared to Nyströmformer, the non-trivial contributions of our proposed method include the identification of the potential issues in Nyströmformer and the proposal of a new algorithm to better utilize the Nyström method. Specifically, by lifting the non-symmetric attention score matrix to a PSD matrix whose off-diagonal block is the target, we improve the approximation performance (Figure 2). As a consequence, our work also connects the studies on efficient transformers to the results on efficient kernel learning with the Nyström method, which can further benefit the design of attention mechanisms, as remarked in Line 330.

---

### Official Review · Reviewer_Dpwz · 2021-07-18

**Rating:** 6
**Confidence:** 2

**Summary:**

The author proposes a modified Nyström method named Skyformer to approximate the Kernelized Attention. They also provide a theoretical guarantee that the approximation error is small in terms of the spectral norm. Extensive experiments are also conducted to show that Skyformer achieves comparable performance to vanilla self-attention with fewer computational costs.

**Limitations And Societal Impact:**

- In Theorem 2, it seems that the sampling size d is independent of the approximate accuracy \epsilon, which is unreasonable.

- For Figure 3, it is not evident that Skyformer is more stable than the other competitors. Does the author have more solid evidence to show that Skyformer is more stable than softmax?

- For Figure 3, It is fairer if the author uses the real-time as X-axis, instead of steps. Also, in Table 2, it seems that Skyformer is not the fastest algorithm.

**Main Review:**

The paper is well written and easy to follow. In this paper, A new variant of the Nyström method, Skyformer, is proposed to estimate the Kernelized Attention. Specifically, they expand the original Kernelized Attention to a larger PSD Kernelized Attention and then use the Nyström method to approximate the larger PSD matrix. The theoretical result shows that the complexity of Skyformer is comparable to the complexity of most other efficient transformers. Experiment results also facilitate the author's claim. Although I admire the theoretical contribution of this work, I am worried about the practice of this method since it enlarges the original dimension n to 2n and needs to calculate the inverse of a matrix.

**Time Spent Reviewing:**

5

---

> ### Author Response · Authors · 2021-08-10
> **Author Response of Paper512**
>
> We thank you for your review and appreciate all your helpful feedback. We hope the following response can address your concerns.
>
> ## The enlarged dimension and the inverse
>
> We remark that the factor $2$ (in $2n$) is not an issue. In the implementation, we only need to construct $Q_S$, $K_S$, and $C_S$ (c.f. Figure 1), which are of size $n$ or $d_S$ ($\ll n$). Moreover, as $d_S$ is small compared to $n$ when $n$ is large, the cost to do an inversion of $C_S$ is acceptable, which is the justification for the Nyström method, Nyströmformer, and our method.
>
>
> ## Sampling size $d$ in Theorem 2
>
> $d$ does depend on $\epsilon$ through the intermediate variable $d_{stat}$. The definition of $d_{stat}$ involves $\lambda$, which is proportional to $\epsilon$. The form of Theorem 2 mainly follows the results given by Musco and Musco (2017), and we will follow your suggestion to revise the statement to make the dependence clearer.
>
> ## Stability
>
> Thank you for pointing this out. We will add more discussions about the stability improvement in the next version.
>
> For our argument about stability, we mainly refer to [Liu et al., 2020a], which identifies that the amplification of small parameter perturbations in the self-attention module is the root cause of  training instability. We take kernelized attention as mitigation since it contains an automatic normalization. We empirically use Figure 3 and Figure 1 in Appendix A to support our claim.
>
> For further analysis we conduct a toy experiment adapting from Figure 4 in [Liu et al., 2020a] to show that in kernelized attention (and Skyformer) the output changes $f(x, W^*) - f(x, W)$ for parameter changes $W^* - W$ is smaller than in self-attention (and its approximation Nyströmformer). This concept involved is somewhat similar to condition number and below we will formalize it as “instability score”.
>
> We show a table of the averaged ratios between the instability scores of kernelized attention (we also add Skyformer and Nyströmformer for reference) and self-attention to conclude our statement about stability. A ratio smaller than $1$ means higher stability compared to self-attention. We follow all the settings in Table 3 except here we only update the model for 20 steps (we limit the number of steps as suggested by Liu et al. [2020a] to make the results of the same step comparable among different models). In step i for each model we compute the instability score
> $$\tau_i = \|f(x_i, W_i) - f(x_i, W_{i-1})\|_F^2 / \|W_i - W_{i-1}\|_F^2, i=1, \dots, 20$$
> where $f()$ gives the embedding after two layers, $x_i$ is the $i$-th input sequence batch, $W_0$ represents the initial parameters, and $W_i$ represents the parameters after step i. For all i we compute the ratio of a certain method’s $\tau_i$ to the $\tau_i$ of self-attention, and finally average the 20 ratios in the table.
>
> |         Model        | Text | ListOps | Retrieval | Pathfinder | Image |
> |----------------------|:----:|:-------:|:---------:|:----------:|:-----:|
> | Kernelized Attention | 0.83 |   0.77  |    0.64   |    0.74    |  0.62 |
> |       Skyformer      | 0.81 |   0.79  |    0.64   |    0.79    |  0.65 |
> |     Nyströmformer    | 1.03 |   1.01  |    0.97   |    0.99    |  1.02 |
>
> As we can see, both kernelized attention and Skyformer consistently have a lower instability score than self-attention, while the instability score of Nyströmformer, an approximation to self-attention, is roughly the same as self-attention in all the tasks.The results support our claim that the proposed kernelized attention can improve stability.
>
> ## Figure 3 X-axis
>
> The X-axis in Figure 3 represents training time. We will clarify that in the next version.
>
>
> ## Table 2 Time
>
> In Table 2 we report the total runtime of each model to train for 50k steps. We argue that considering the accuracy growth rate, it can be observed in all the tasks Skyformer is one of the fastest. We will also add the figure of validation accuracy changes with respect to training time to the appendix to enhance our statement.

---

### Official Review · Reviewer_oeCF · 2021-07-22

**Rating:** 9
**Confidence:** 4

**Summary:**

The paper presents an alternative approach to attention in the Transformer. There are two modeling contributions:

1. Kernelized attention that omits the typical normalization term used for softmax attention.
2. A Nystrom method approach that allows efficiently approximating the output of the attention mechanism

The contributions also include theoretical results regarding the accuracy of the Nystrom approximation, and experimental results showing better convergence, higher accuracy, and medium efficiency gains from the approach.

Below I summarize my understanding/takeaways from the two key methods; please correct me in the author response if these are inaccurate:
1. As I understand it, "kernelized attention" mean that target keys don't compete with each other, the only thing that matters is their distance to the query. In standard softmax attention, suppose that all key vectors are orthogonal to a query vector -- in that case, attention would weigh all values uniformly. In kernelized attention, on the other hand, if key vectors are all far away from the query vector, then all value vectors would be attended to with near-zero weight.
2. The Nystrom approximation involves sampling some queries and keys, and then computing kernels between all pairs. But whereas Nystromformer computes similarity for query+key pairs only, the proposed approach also computes kernels for query+query and key+key pairs as well.

**Limitations And Societal Impact:**

Limitations and impact are adequately addressed.

**Main Review:**

Overall this is a very solid paper. The methods presented are novel in the context of efficient attention, and they are justified from both a theoretical and experimental standpoint. I particularly appreciate that the error caused by the Nystrom approximation rapidly shrinks as the number of features increases (as shown in Figure 2), and I believe that it should approach zero as the number of features grows to match the length of the sequence. The paper is clear and well-structured, with some minor exceptions noted below. The overall research program of constructing efficient attention variants has gained a lot of interest in the community, and this paper appears to represent significant progress on this research direction.

The paper offers two contributions that are in principle orthogonal: kernelized attention, and the Nystrom approximation. Indeed, kernelized attention (without any approximations) is presented as a standalone ablation. Why about evaluating the Nystrom approximation in isolation? I believe this would just involve taking the proposed Skyformer and re-introducing a $D^{-1}$ normalization term. Does this make sense conceptually, and have the authors tried it in practice?

In terms of writing, the only point where I was confused was Theorem 2, which I still don't understand. Line 222 simultaneously introduces a new scalar lambda, uses a matrix C never previously mentioned in the theorem, and *also* introduces a new concept C-bar. There is just too much happening all at once. It would help to introduce all terms one by one, specify in words what lambda is, and otherwise clarify.

I have some questions regarding the LRA evaluations. First, is there a reason the Image task is omitted? Without an explanation in the text it's possible to interpret this as cherry-picking. Next, Retrieval scores are a full 20 points higher than the LRA paper. Nystromformer has similar results, so I trust that the results presented are correct, but it would be helpful to have a comment in the paper regarding this situation. Finally, Pathfinder scores are actually *lower* than both Tay et al (LRA) and Xiong et al. (Nystromformer). Do you know why this might be?

Overall LRA benchmarking is becoming a bit of a mess, because results from two different papers can't be compared against each other. This situation is not limited to only the present paper, but it's still disappointing because it makes it possible for individual papers to use cherry-picked baselines or poor hyperparameter choices for competing methods, without no easy way to flag it using cross-paper comparisons. I understand that it's hard for any one paper to address systemic benchmarking issues, but the results would certainly be stronger if the authors had a way to mitigate these concerns. If anything a little more discussion to justify the evaluation would help.

Minor
- Why the name Skyformer? I'm just curious, because the naming choice is never discussed in the paper.
- line 27: I would not characterize Reformer as a low-rank approximation, since it can learn full-rank sparsity patterns
- line 77: "By reduce" is ungrammatical
- Equation on line 177: there is no V term on the right side of the equality, so I think this equation isn't quite right
- Figure 1: $K_S^T$ should have shape $d_S$ x $n$

**Time Spent Reviewing:**

7

---

> ### Author Response · Authors · 2021-08-10
> **Author Response of Paper512**
>
> We thank you for your acceptance of the paper and appreciate all your valuable comments.
>
> ## Nyström approximation in isolation
>
> For the evaluation of the Nyström method alone, we've done some exploratory experiments. We find it suffers a more severe gradient explosion issue than usual transformers. We speculate that it is because the middle matrix $C_S$ (in Figure 1) inherits the high condition number of the original attention score matrices (Line 38), while the derivative of matrix inverse ($(A^{-1})’ = - A^{-1} A’ A^{-1}$) has an even larger condition number and thus leads to exploding gradients during backpropagation. We will add the discussion above to the next version of this paper.
>
> ## Writing of Theorem 2
>
> The matrix $C$ in Theorem 2 comes from the definition in Line 190. We agree that the new variables introduced in Theorem 2 have not been fully explained. We will take your advice to rewrite the statement  to improve its readability and clarity.
>
> ## LRA image
>
> We did not report the performance on the image task because we failed to reproduce the results (all models have significantly lower accuracies) in Xiong et al. (Nyströmformer). Recently we have fixed some errors in their codebase and searched for appropriate configurations to reproduce their results. The results are reported in the following table.
>
> | Model                |  Text | ListOps | Retrieval | Pathfinder | Image |  AVG. |
> |----------------------|:-----:|:-------:|:---------:|:----------:|:-----:|:-----:|
> | Self-Attention       | 61.95 |  38.37  |   80.69   |    65.26   | 40.57 | 57.37 |
> | Kernelized Attention | 60.22 |  38.78  |   81.77   |    70.73   | 41.29 | 58.56 |
> | Nyströmformer        | 64.83 |  38.51  |   80.52   |    69.48   | 41.30 | 58.93 |
> | Linformer            | 58.93 |  37.45  |   78.19   |    60.93   | 37.96 | 54.69 |
> | Informer             | 62.64 |  32.53  |   77.57   |    57.83   | 38.10 | 53.73 |
> | Performer            | 64.19 |  38.02  |   80.04   |    66.30   | 41.43 | 58.00 |
> | Reformer             | 62.93 |  37.68  |   78.99   |    66.49   | 48.87 | 58.99 |
> | Bigbird              | 63.86 |  39.25  |   80.28   |    68.72   | 43.16 | 59.05 |
> | Skyformer            | 64.70 |  38.69  |   82.06   |    70.73   | 40.77 | 59.39 |
>
> Similar to the results in Xiong et al. (Nyströmformer), Reformer has significantly better performance than other x-formers. During training, Reformer keeps decreasing the validation loss while most other x-formers converge quickly (roughly at the 5k step out of 35k step). Skyformer has slightly worse performance than Reformer in image classification but still comparable to other models.
>
>
> ## LRA performance difference
>
>
> First we would like to state that we do not follow every setting in Nyströmformer due to the hardware limitation. Our V100 GPU has 16 GB memory, and therefore we reduce the approximation dimension ($d_S$ in our model, random feature dimension in Performer, etc.) from 256 to 128, and in some certain tasks we have to leverage gradient accumulations so as to allow using the same batch sizes as the ones in Xiong et al. (Nyströmformer). Those compromises might cause slight differences.
>
> For the lower performance in pathfinder, we speculate about the possible reasons as follows. Tay et al. (LRA) extensively search the optimal architecture for each model in a task (so that even the number of layers and the number of heads can be different for the models in the same task), while Xiong et al. (Nyströmformer) and we fix the same architecture for different models / tasks. For the gap between the results in Xiong et al. (Nyströmformer) and ours, we remark in our experiments that it is related to instability: we do observe that sometimes some models can have accuracy similar to the ones reported in Xiong et al. (Nyströmformer), while in the other runs they have poor performance due to the convergence to bad local minima.
>
>
> ## LRA cross-paper comparisons
>
> For your concern regarding the cross-paper comparisons, we agree with your suggestion that we can do more than simply following previous work. Although we may not be able to do the extensive parameter search as Tay et al. (LRA) do, we can report more detailed experimental results with multiple sets of hyperparameters as well as significance testing to strengthen the robustness of our conclusion.

---

> > ### Comment · Reviewer_oeCF · 2021-08-30
> > **Re: Author response**
> >
> > Thank you for your response!
> >
> > The comments are quite helpful in addressing my questions. This information and new experimental results should also be included when revising the paper.
> >
> > It appears that my score is quite different from the other reviewers; nevertheless, I stand by my evaluation. I think that resolving the quadratic time complexity of attention is a very important problem to work on, especially now that huge pre-trained Transformers have become the go-to approach not only in NLP, but also in other domains including computer vision. The paper is extremely clear and well-written, the proposed method is quite straightforward to implement by following the text of the paper, and experimental results are quite good.
> >
> > While other reviewers find Nyströmformer to be quite similar, my personal evaluation is that only the use of the Nystrom method is in common between the two, while the actual details of the approaches are quite different. The difference is also not only in how the Nystrom method is applied, but also in how normalization (e.g. the softmax in regular attention) is handled. The present paper presents what I see as a new approach, with clear motivation, without any "hacks" to make it work, and with theoretical backing for the validity of the approach.

---

### Official Review · Reviewer_5adD · 2021-07-23

**Rating:** 6
**Confidence:** 4

**Summary:**

The paper proposes a method (Skyformer) to apply Nystrom approximation to the attention matrix. It embeds the attention matrix inside a larger PSD matrix (unlike Nystromformer), allowing Nystrom method to work well. Theoretical and empirical validation shows that the method is competitive with other forms of efficient attention (e.g. BigBird, Performer, Reformer).

**Limitations And Societal Impact:**

Yes

**Main Review:**

Strengths:
1. The paper is well-written and easy to understand. Section 4 provides valuable intuition, and I enjoyed reading it.
2. The method can significantly speed up attention computation, which is necessary for scaling to longer sequences.
3. The method is novel to the best of my knowledge.

Weaknesses:
1. Limitation of application: it's not clear to me how widely the proposed method can be apply, as there's no discussion of its limitations. I'm not sure which conditions will allow it to work well.
2. The empirical validation is not sufficient. As the paper proposes a new variant of attention (based on Gaussian kernel), one has to demonstrate this works on different tasks (e.g. language modeling, translation, etc.), not just classification as in the long-range arena benchmark.

Questions:
1. Section 4.2, instead of the block matrix [[phi(Q, Q), phi(Q, K)], [phi(K, Q), phi(K, K)]], would it also work to embed phi(Q, K) in the block matrix [[0, phi(Q, K)], [phi(K, Q), 0]]? I think this matrix is also PSD, and it has a lot of zeros so it might save some computation.
2. Figure 2: (a) Isn't Skyformer approximating a different matrix (the matrix C instead of the attention matrix)? Why would Skyformer have low approximation error wrt the attention matrix here?
(b) Maybe Performer is having problem because it also tries to approximate the normalization constant D^{-1}, which Skyformer does not do. Would that make this an unfair comparison, because Skyformer is approximating a different matrix and Performer is doing more work to approximate the denominator?

Comments:
- Line 129, (D_Q)_ii = ||q_i||^2 / sqrt(p) should be (D_Q)_ii = exp(-||q_i||^2 / sqrt(p)).
- Equation after line 177 equation, this definition is for the matrix C, not for Kernelized-Attention(Q, K, V). I assume kernelized-attention is the product of C and V.
- Line 178, (D_Q)_ii = ||q_i||^2 / sqrt(p) should be (D_Q)_ii = exp(-||q_i||^2 / sqrt(p)).

============== Post-rebuttal

Thank you for the response to my questions. That has helped clarified some of my misunderstanding of the method.

I would still encourage the authors to validate the method on a wider variety of applications to make a stronger case.

I've increased my rating.

**Time Spent Reviewing:**

3

---

> ### Author Response · Authors · 2021-08-10
> **Author Response of Paper512**
>
> We thank you for the precious comments and updates.
>
> ## W1. Limitation of application
>
> The usage of Skyformer might be limited to long-sequence tasks since for small sequence length $n$ the statistical dimension $d_{stat}$ (defined in Line 225) might be even close to $n$.
>
> To make the claim above clear, we first reiterate the efficiency of Skyformer is related to $d_{stat}$. As implied by Theorem 2, the intrinsic difficulty of approximating a raw attention score matrix is concluded as $d_{stat}$, which corresponds to the effective rank of matrix $\bar{C}$. For Skyformer, its complexity depends on the sub-sample size $d$ (the size of the sub-sampling matrix $S$), and a large $d_{stat}$ leads to a large $d$ (c.f. Theorem 2), making the Nyström method inefficient as we imply in Line 233.
>
> The classical theory for statistical dimension only guarantees that $d_{stat}$ is small (compared to $n$) when $n$ is large enough, and it is possible the statistical dimension associated with a short sequence might be even close to the sequence length $n$. A large $n$ thus serves as a condition to make the method work. From Figure 2 we observe that our method performs better with larger $n$’s. We will add the discussion into the next version.
>
>
>
>
>
> ## W2. Empirical validation other than LRA
>
> We use LRA (currently the only standard long sequence modeling benchmark for long transformers) to evaluate our models for now because we are focusing on the long-sequence complexity problem instead of proposing new attention modules for general usage. Since LRA covers multiple long-sequence modeling tasks of different difficulty levels, we consider LRA to be sufficient for evaluating long transformers.
>
> ## Q1 about Section 4.2
> The new block matrix $[[0, \phi(Q, K)], [\phi(K, Q), 0]]$ is not PSD unless $\phi(K, Q)=0$. For example, consider the 2-d example of  $[[0, a], [a, 0]]$ for any nonzero $a$.
>
> ## Q2 about Figure 2
>
>
> In Figure 2, all methods are approximating the same target, the original self-attention output.
> The label “Skyformer” here means that we use the technique behind Skyformer, Eq. (5), to approximate the raw attention score matrix $A$ in self-attention. We apologize for the confusion. We aim to clarify that in Figure 2, Skyformer also needs to first approximate $A$, and then approximate $D$, as Performer does.
>
> For Performer, we think this random-feature-based method works as expected since the random feature method are known to approximate gram matrices worse than the Nyström method for most common kernels [1], [2, Section 1].
>
> [1] Yang, Tianbao, et al. "Nyström method vs random fourier features: A theoretical and empirical comparison." Advances in neural information processing systems 25 (2012): 476-484.
> [2] Avron, Haim, Kenneth L. Clarkson, and David P. Woodruff. "Faster kernel ridge regression using sketching and preconditioning." SIAM Journal on Matrix Analysis and Applications 38.4 (2017): 1116-1138.

---

### Official Review · Reviewer_ugqB · 2021-07-24

**Rating:** 5
**Confidence:** 4

**Summary:**

This paper proposes Skyformer a new xformer variant that uses Gaussian kernels and the Nystorm method.

**Main Review:**

The idea does sound interesting and novel, the experiments on long range arena look satisfactory.

Results look borderline but overall better than some of the existing baselines while being faster/less memory hungry.

The improvements of Nystromformer is that: "Nyströmformer blindly applies the Nyström method to a non102 PSD matrix, and thus fails to ultize the full potential of Nyström method." the authors solve this by "instead lifting the kernelized attention score matrix into a large PSD
matrix which contains the target non-PSD matrix as its off-diagonal block."

I have some minor concerns.

1) I find that the intro is an over nag. The paper starts off by telling us how expensive GPT-3 training is (or how long it takes). I don't think the proposed method solves this problem. I would advise the authors to keep the exposition short. Most reviewers in the space know the problems with Transformers and the quadratic complexity problem.
2) What happened to the image CIFAR task on LRA? Why was that omitted?
3) A nit but why is the model called Skyformer? What does SKY mean here?


**Time Spent Reviewing:**

1

---

> ### Author Response · Authors · 2021-08-10
> **Author Response of Paper512**
>
> We appreciate your helpful feedback. We hope the following answers will address your concerns.
>
> ## Q1. Introduction
>
> Thank you for the advice. We will revise the introduction and make it more concise.
>
> ## Q2. LRA image
>
> We did not report the performance on the image task in submission because we failed to reproduce the results (all models have significantly lower accuracies) in Xiong et al. (Nyströmformer) at that time. Recently we have fixed some errors in their codebase and searched for appropriate configurations to reproduce results similar to theirs. The results are reported in the following table.
>
> | Model                |  Text | ListOps | Retrieval | Pathfinder | Image |  AVG. |
> |----------------------|:-----:|:-------:|:---------:|:----------:|:-----:|:-----:|
> | Self-Attention       | 61.95 |  38.37  |   80.69   |    65.26   | 40.57 | 57.37 |
> | Kernelized Attention | 60.22 |  38.78  |   81.77   |    70.73   | 41.29 | 58.56 |
> | Nyströmformer        | 64.83 |  38.51  |   80.52   |    69.48   | 41.30 | 58.93 |
> | Linformer            | 58.93 |  37.45  |   78.19   |    60.93   | 37.96 | 54.69 |
> | Informer             | 62.64 |  32.53  |   77.57   |    57.83   | 38.10 | 53.73 |
> | Performer            | 64.19 |  38.02  |   80.04   |    66.30   | 41.43 | 58.00 |
> | Reformer             | 62.93 |  37.68  |   78.99   |    66.49   | 48.87 | 58.99 |
> | Bigbird              | 63.86 |  39.25  |   80.28   |    68.72   | 43.16 | 59.05 |
> | Skyformer            | 64.70 |  38.69  |   82.06   |    70.73   | 40.77 | 59.39 |
>
>
> Similar to the results in Xiong et al. (Nyströmformer), Reformer has significantly better performance than other x-formers. During training, Reformer keeps decreasing the validation loss while most other x-formers converge quickly (roughly at the 5k step out of 35k step). Skyformer has worse performance than Reformer in image classification but is still comparable to other models.
>
>
> ## Q3. Skyformer
>
> For S, K, and Y we mean “Symmetrization”, “Kernel”, and “NYström”. We will add this explanation to the introduction.

---

### Author Response · Authors · 2021-08-10
**Author Response of Paper512**

Dear Area Chair and Reviewers,

We appreciate all the reviewers for their valuable comments. We hope our responses address the reviewers’ concerns. We are also ready for any future questions in the rolling discussion.

---

> ### Comment · Reviewer_4vy6 · 2021-08-30
> **Response**
>
> I read through all the reviews and responses, and I am happy with the original score I gave.

---

> ### Author Response · Authors · 2021-09-02
> **Author Response of Paper512**
>
> Dear Area Chair and Reviewers,
>
> We are glad that our previous response more or less help clarify some points, and sincerely grateful for the reviewers' time to read our long response. We treasure the feedback from the reviewers, which helps us notice the flaw and improve the quality of this work. If there are still some concerns from the reviewers, we will keep on striving to address them.

---

### Decision · Program_Chairs · 2021-09-27

**Decision:**

Accept (Poster)

**Comment:**

The reviewers agree that this is a solid paper re-introducing the Nystrom method for Transformers to address the quadratic space and time complexity of the regular attention module. Thus, it leverages the fruitful area of research on using kernel approaches to improve regular Transformers. The novelty given that Nystrom method was previously used in that context is limited, but the empirical results are strong and furthermore the authors explain in detail approximation guarantees of their method. What is missing is equally detailed analysis of the computational time of the presented method. Nystrom algorithm is in general expensive, but the authors manage to bypass some of its complexities, as showed in the experimental section. More detailed discussion regarding Nystrom method in theory and practice would strengthen the paper.